# Detecting variations in ovulation and menstruation during the COVID-19 pandemic, using real-world mobile app data

Brian T. Nguyen[1]*, Raina D. Pang[2], Anita L. Nelson[3], Jack T. Pearson[4], Eleonora Benhar Noccioli[4], Hana R. Reissner[1], Anita Kraker von Schwarzenfeld[4], Juan Acuna[5,6]

1 Department of Obstetrics and Gynecology, Section of Family Planning, Keck School of Medicine of the University of Southern California, Los Angeles, CA, United States of America, 2 Department of Preventive Medicine, Keck School of Medicine of the University of Southern California, Los Angeles, CA, United States of America, 3 Department of Obstetrics and Gynecology, Western University of Health Sciences, Pomona, CA, United States of America, 4 Natural Cycles, Stockholm, Sweden, 5 Department of Epidemiology and Public Health, Khalifa University, Abu Dhabi, United Arab Emirates, 6 Research and Data Intelligence Support Center, Khalifa University, Abu Dhabi, United Arab Emirates

* nguyenbt@usc.edu

**Data Availability Statement:** The data underlying this study are publicly available at doi.org/10.5061/dryad.0cfxpnw2k.

## Abstract

### Background

As war and famine are population level stressors that have been historically linked to menstrual cycle abnormalities, we hypothesized that the COVID-19 pandemic could similarly affect ovulation and menstruation among women.

### Methodology

We conducted a retrospective cohort study examining changes in ovulation and menstruation among women using the Natural Cycles mobile tracking app. We compared de-identified cycle data from March-September 2019 (pre-pandemic) versus March-September 2020 (during pandemic) to determine differences in the proportion of users experiencing anovulation, abnormal cycle length, and prolonged menses, as well as population level changes in these parameters, while controlling for user-reported stress during the pandemic.

### Findings

We analyzed data from 214,426 cycles from 18,076 app users, primarily from Great Britain (29.3%) and the United States (22.6%). The average user was 33 years of age; most held at least a university degree (79.9%). Nearly half (45.4%) reported more pandemic-related stress. Changes in average cycle and menstruation lengths were not clinically significant, remaining at 29 and 4 days, respectively. Approximately 7.7% and 19.5% of users recorded more anovulatory cycles and abnormal cycle lengths during the pandemic, respectively. Contrary to expectation, 9.6% and 19.6% recorded fewer anovulatory cycles and abnormal cycle lengths, respectively. Women self-reporting more (32.0%) and markedly more (13.6%) stress during the pandemic were not more likely to experience cycle abnormalities.

**Funding:** This is an investigator-initiated study by BTN, HRR, and ALN who have no commercial affiliation with Natural Cycles. While JA is an advisor for Natural Cycles, he has not received any salary or financial support for participation in this research. Natural Cycles provided support in the form of salaries for authors JTP, EBN, and AKvS, but did not have any additional role in the study design, data collection and analysis, decision to publish, or preparation of the manuscript. The specific roles of these authors are articulated in the "author contributions" section.

**Competing interests:** I have read the journal's policy and the authors of this manuscript have the following competing interests: JTP, AKvS, & EBN are affiliated with Natural Cycles, which provided the data for this study. JA is a medical advisor to Natural Cycles.

## Conclusions

The COVD-19 pandemic did not induce population-level changes to ovulation and menstruation among women using a mobile app to track menstrual cycles and predict ovulation. While some women experienced abnormalities during the pandemic, this proportion was smaller than that observed prior to the pandemic. As most app users in this study were well-educated women over the age of 30 years, and from high-income countries, their experience of the COVID-19 pandemic might differ in ways that limit the generalizability of these findings.

## Introduction

On March 11, 2020 the World Health Organization (WHO) declared the COVID-19 outbreak a global pandemic, calling upon all countries to take urgent and aggressive action to prevent further spread of the disease. The pandemic declaration led to increasingly strict and widespread stay-at-home orders, which contributed to population level concerns about the risk of not only infection and loss of life, but the potential irrecoverable loss of livelihood, as non-essential businesses were shut down and citizens deprived of income, while facing continued expenses. Persisting beyond a full year, the COVID-19 pandemic produced a global population-level stressor that continues to influence how people live.

Historically, population level stressors have been linked to menstrual changes. Examples include World War II from 1939–1945 [1], the Dutch Famine from 1944–1955 [2], and the Desert Storm War in 1996 [3]. These examples represent a range of stressors, whether they be physical vs. psychological, direct vs. indirect, or short vs. long-term. At the extremes of stress, a case series of women studied prior to being executed noted that almost all became amenorrheic [4]. In an observational study of Lebanese women exposed to wartime bombing from April 11–27, 1996, 35% experienced menstrual aberrations for 3 months, as compared to an unexposed group that reported abnormalities in 2.6% [5]. Abnormal menstrual patterns are also commonly reported among desert-dwelling hunter–gatherer women who face difficult living conditions [6]. Yet mortal stresses of this magnitude are not required to induce menstrual aberrations; chronic job-related stress can be a contributor [7, 8]. Further, in a study of Japanese college students, 15.8% reported a correlation between their school examinations and irregular menses [9]. The range of stressors that can influence one's likelihood of menstrual irregularity is wide, suggesting that other mediating factors may play a role.

The impact of stress on the reproductive system is grounded in biology whereby stress-related glucocorticoids can inhibit the release of gonadotropin releasing hormone (GnRH), luteinizing hormone (LH), and estradiol (E2) from the hypothalamic-pituitary-gonadal axis, which is most easily observed as secondary amenorrhea. More subtle manifestations would include delays in ovulation, anovulation, and changes in both cycle and menstruation lengths.

The COVID-19 pandemic represents a unique stressor, independent of whether individuals were infected by the virus, because of its far-reaching psychologic, social, and economic consequences beyond physical alone. Consequently, reproductive age females may have experienced changes in their ovulatory and menstrual cycles during this time. No studies have yet examined the influence of stress linked to an ongoing global pandemic on ovulatory and menstrual changes among a reproductive age female population. The pandemic provides an opportunity to identify and characterize potential changes.

Natural Cycles is the first Food and Drug Administration (FDA) cleared and CE-marked mobile app that uses women's records of menstruation and basal body temperatures to identify

their fertile window and estimate appropriate times when they can have unprotected sexual intercourse and be reasonably certain that they will avoid pregnancy. As of the beginning of the COVID-19 pandemic in 2020, Natural Cycles had more than 1.5 million members using the app for contraception, making its database one of the largest collections of menstrual cycle data ever compiled. We conducted analyses using aggregated, real-world menstrual cycle data from app users both before and during the COVID-19 pandemic with the primary objective of detecting potential changes in ovulation, cycle length, and menstrual duration during the pandemic. The secondary objective of this study was to examine the association of any menstrual changes with self-reported perceived stress related to the pandemic.

## Materials and methods

### Data collection

Our data was comprised of real-world, sociodemographic, limited clinical, and menstrual cycle data submitted by users of the Natural Cycles pregnancy prevention and fertility tracking mobile application [10]. All users who contributed data agreed to make their data available for clinical investigation prior to starting application use. All data was de-identified by the Natural Cycles data management team (EBN) and stored within a closed database, which was transferred to the research team at the University of Southern California (USC; BTN, RDP, HR, ALN) for subsequent data analysis. This research plan was reviewed and approved by the USC Institutional Review Board, which classified the study as exempt, non-human subjects research (HS-20-00402).

### Inclusion and exclusion criteria

We included users who registered an account for contraception prior to September of 2019 and who consented to release their de-identified data for research purposes. We excluded cycle data from users who were breastfeeding, who reported a pregnancy in the 12 months prior to registration within the data collection period, and who became pregnant during use of the app. We additionally excluded any user on a hormonal contraceptive method, as well as reporting at the time of registration, a diagnosed medical condition that could influence cycle regularity, such as polycystic ovarian syndrome, endometriosis, thyroid disorders, and peri-menopausal symptoms. To account for any undiagnosed secondary amenorrhea or ovulatory disorders, as well as to improve our ability to detect pandemic-induced changes in ovulation and menstruation, we excluded users who recorded any cycle length lasting more than 90 days during the pre-COVID analysis frame. We further excluded users who did not contribute at least two cycles of data prior to the COVID-19 pandemic and at least two cycles during the COVID-19 data collection period.

With respect to cycle data, we included only complete cycles (i.e., cycles with a start and an end) collected from March through September 2019 (pre-COVID) and March through September 2020 (during COVID). All cycles at the beginning of the sampling frame started in March; all cycles included from the end of the sampling frame must have started in September even if the cycles did not end until later. We excluded cycle data starting from October 2019 through February 2020 due to ambiguity at the time about the scale and threat of infection among various populations of users. We did not include cycles in which the Natural Cycles algorithm did not have enough data to definitively determine whether an ovulation occurred within the cycle (i.e., not enough user data entered). To improve the precision of contributed cycle data, we further excluded cycles that were not validated by at least 10 basal body temperature entries.

## Outcomes of interest

We were primarily interested in examining the proportion of users experiencing anovulation, abnormal cycle length, and prolonged menstruation among app users pre-COVID compared to during COVID. Anovulatory cycles were defined according to the Natural Cycles app's proprietary algorithm. The algorithm identifies ovulation retrospectively based on the first day of menstruation and basal body temperatures, which may be supplemented by positive urinary LH tests. Basal body temperatures are recorded each morning using a thermometer sensitive to the hundredth place, and with measures excluded if the user reports any illness, alcohol intake, or changes in sleep that might influence basal temperatures. Users of Natural Cycles record basal temperatures for approximately 70% of the days; approximately 25% use LH tests. To reduce the risk of misidentifying ovulations, the algorithm reports ovulation by rising basal body temperature only if the average temperature from three consecutive calendar days is greater than the woman's follicular phase average and her baseline average across all data entries, as well as consistent with her luteal phase average [11]. If no temperature rise is observed and the data quality and quantity is deemed sufficient, the cycle is flagged as anovulatory. Of note, users with stable measurements (e.g., small day-to-day variations in the same cycle phase), require fewer data points for the Natural Cycles algorithm to draw conclusions about changes in the basal body temperature. Cycles with low data quality (e.g., high day-to-day temperature variability) or data that is insufficient to detect or exclude an increase in the basal body temperature are excluded by the analysis. While missing data may affect the app's ability to predict the exact day of ovulation, they are not expected to affect its determination of whether ovulation has occurred. Based on criteria from the International Federation of Gynecology and Obstetrics (FIGO), we defined abnormal cycle length as lasting less than 24 days or more than 38 days, and prolonged menstruation length as lasting greater than 8 days [12, 13].

## Covariates

Natural Cycles implemented a query on May 11, 2020 of its users' experience of pandemic-related stress. Users could provide a response to this item through June 11, 2020. We incorporated this measure to examine a stress-related pathway for menstrual abnormalities. The researchers asked users to rate two Likert-type items: (1) "Thinking about your stress level before the COVID-19 pandemic started, how stressed were you then?" (2) "How stressful is the COVID-19 pandemic to you now?" The Likert items were anchored from "Not at all stressed" (1) to "Extremely stressed" (5). We then calculated a COVID-related stress change score by subtracting the user's report of stress before COVID from their stress rating during COVID. For ease of interpretation, we categorized perceived changes in stress using five categories: "much less (1)," "less (2), "unchanged (3)," "more (4)," and "much more (5)." In addition, we used sociodemographic data and reproductive histories provided by users at the time of app registration and updated during use of the app as covariates in our analysis. These covariates included: age, country of registration, education (from less than a high school degree to graduate degree), relationship status (e.g., in a relationship, engaged or married, it's complicated, and single), history of pregnancy, and whether they have any children.

## Data analysis

At the population level, we examined differences in average cycle lengths and menstruation lengths before and during the COVID-19 pandemic via paired t-tests. We calculated the proportions of users experiencing an anovulatory cycle before and during the pandemic and analyzed for differences via Chi-square tests of association. Given that individual users did not contribute the same number of cycles to the study, we compared differences in the proportions

of abnormal menstrual parameters recorded by calculating for each user their total number of cycles with anovulation, abnormal cycle length, and prolonged menses for each sampling frame and divided them by the number of cycles contributed. We examined for statistically significant changes in these abnormal cycle parameters, pre-COVID and during COVID, via paired t-test.

Given our interest in COVID-related stress as a potential mediator of abnormal cycles, we conducted a subset analysis of data from individuals contributing stress ratings during the pandemic and examined its relationship categorically via Chi-square tests and continuously via paired t-tests. We examined the role of the user's age, country of registration, education, relationship status, and history of pregnancy and/or children on any increase in mean anovulation, cycle length, and menstruation length among users via Chi-square tests. We included covariates associated with increased anovulatory cycles, abnormal cycle lengths, and abnormal menstruation length (at an alpha level of 0.05) in a separate multivariable logistic regression to determine each covariate's adjusted odds of influencing the outcome of interest. All analyses were conducted in Stata (Version SE/14.2; College Station, TX).

## Results

Our sampling frame included a total of 214,426 cycles of data from 18,076 individual users from over 60 countries worldwide (Table 1). Users were primarily from Great Britain (29.3%), the United States (22.6%), and Sweden (17.8%). The average user was 32.5 ± 5.8 years of age at the time of analysis, and most held at least a university degree (79.9%). Most users reported being in a relationship, engaged, or married (85.1%); only 25.6% of users reported ever being pregnant and 16.9% of users reported having at least one child.

The proportion of those reporting being very to extremely stressed rose from 46.2% pre-COVID to 61.1% during COVID. Nearly half of users (45.4%) reported more pandemic-related stress, with 33.2% reporting no change, and 21.4% reporting less stress compared to the pre-COVID period. Individuals between the ages of 25–34 years, from the United States, with a graduate level degree, who reported being engaged or married, and having a child were significantly more likely to report more stress during the pandemic than prior.

Users individually contributed approximately 6 cycles to each of the pre-COVID and during COVID sampling frames. With respect to cycle characteristics (Table 2), the average cycle length among users significantly decreased from 29.40 (95%CI 29.34–29.46) days pre-COVID to 29.16 (95%CI 29.10–29.22) days during COVID (p<0.001). The average menstrual duration significantly increased from 4.21 (95%CI 4.19–4.23) days to 4.32 (95%CI 4.30–4.34) days. The average incidence of anovulation and abnormal cycle length decreased significantly across cycles, from 2.9% (2.7%-3.0%) to 2.5% (2.3%-2.6%) and 8.7% (8.4%-8.9%) to 8.0% (7.8%-8.2%), respectively, while the average incidence of prolonged menstruations increased from 0.9% (0.8%-1.0%) to 1.0% (9.0%-1.1%).

At the user level, the proportion of users experiencing any anovulatory cycles and any abnormal cycle lengths decreased from 11.1% to 9.1% (p<0.001) and 28.4% to 26.6% (p<0.001), respectively. The proportion experiencing increased menstrual duration increased from 3.5% to 3.9% (p = 0.04). We did not detect any statistically significant associations between changes in user ratings of stress before and during the pandemic and abnormal cycle parameters (Fig 1, Table 3).

To explore factors associated with changes in the proportion of anovulatory cycles and abnormal cycle lengths after the pandemic, we conducted a multinomial logistic regression, setting no changes in the proportion of abnormalities as the reference point. We included those potential covariates significantly associated with changes in abnormal cycle parameters

**Table 1. Characteristics of natural cycles users contributing at least two cycles of data prior to and during the COVID-19 pandemic (n = 18,076 users; N = 214,426 cycles).**

| | Change in perceived stress from pre-COVID to during COVID pandemic, N = 10,294 | | | Total users, N = 18,076 |
| --- | --- | --- | --- | --- |
| | Less or much less, n = 2206 | Unchanged, n = 3419 | More or much more, n = 4669 | n (%) |
| Age (%) [mean 32.5 ± 5.8 years] *** | | | | |
| <25 | 103 (21.4) | 176 (36.6) | 202 (42.0) | 911 (5.0) |
| 25–34 | 1479 (22.2) | 2104 (31.6) | 3071 (46.2) | 11552 (63.9) |
| 35–44 | 544 (19.9) | 980 (35.8) | 1215 (44.4) | 4861 (26.9)) |
| 45+ | 80 (19.1) | 159 (47.9) | 181 (43.1) | 752 (4.2) |
| Country (%) *** | | | | |
| Great Britain | 700 (22.5) | 942 (30.2) | 1475 (47.3) | 5293 (29.3) |
| United States | 436 (18.6) | 717 (30.6) | 1189 (50.8) | 4091 (22.6) |
| Sweden | 421 (22.3) | 742 (39.3) | 726 (38.4) | 3223 (17.8) |
| Other | 649 (22.0) | 1018 (34.6) | 1279 (43.4) | 5469 (30.3) |
| Education (%) ** † | | | | |
| High school or less | 219 (21.7) | 385 (38.1) | 407 (40.3) | 1672 (11.2) |
| Vocational training | 178 (21.3) | 280 (33.5) | 378 (45.2) | 1322 (8.9) |
| University degree | 1377 (21.1) | 2139 (32.8) | 3014 (46.2) | 10701 (71.8) |
| PhD degree | 156 (21.5) | 215 (29.6) | 355 (48.9) | 1208 (8.1) |
| Relationship status (%) **† | | | | |
| In a relationship | 1092 (22.2) | 1625 (33.1) | 2198 (44.7) | 8211 (52.6) |
| Engaged or married | 611 (19.5) | 1046 (33.3) | 1483 (47.2) | 5071 (32.5) |
| Single | 232 (22.1) | 367 (35.0) | 451 (43.0) | 1700 (10.9) |
| It's complicated | 109 (27.1) | 119 (29.6) | 174 (43.3) | 630 (4.0) |
| Pregnancies (%) † | | | | |
| None | 1507 (21.8) | 2241 (32.4) | 3169 (45.8) | 11479 (74.4) |
| At least one | 490 (20.4) | 842 (35.0) | 1074 (44.6) | 3949 (25.6) |
| Children (%) * † | | | | |
| None | 1747 (21.8) | 2634 (32.9) | 3637 (45.4) | 13255 (83.2) |
| At least one | 308 (19.2) | 563 (35.1) | 731 (45.6) | 2686 (16.9) |

*p<0.05

**p<0.01

***p<0.001.

†Sum different from total number of users due to missing data.

at the p<0.05 level. While changes in stress were not linked to changes in cycle parameters, we included this covariate given our interest in the potential role of subjective stress on menstrual changes (Table 4). Via this regression, only age and relationship status remained independently associated with any of the cycle outcomes. Users above the age of 45 years were more likely to report more anovulatory cycles, abnormal cycle lengths, and prolonged menses. Individuals who were in a relationship or were engaged or married were less likely than their complicated and single counterparts to experience more anovulatory cycles.

## Discussion

The COVID-19 pandemic created an environment of global stress and an unprecedented opportunity to explore the influence of a potential chronic stressor on menstrual cycle parameters. No studies to date have analyzed large-scale, daily self-reported and biologically verified menstrual cycle data during a pandemic. Based on observations from previous historical

**Table 2. Menstrual characteristics and abnormalities prior to and during the COVID-19 pandemic (n = 18,076 users; N = 214,426 cycles).**

| | | | Pre-COVID: Mar-Sep 2019 (n = 108,021 cycles) | During COVID: Mar-Sep 2020 (n = 106,405 cycles) | p-value |
|---|---|---|---|---|---|
| | | | Mean (95%CI) | Mean (95%CI) | |
| Cycle length | | | 29.40 (29.34–29.46) | 29.16 (29.10–29.22) | <0.001 |
| Menstrual duration | | | 4.21 (4.19–4.23) | 4.32 (4.30–4.34) | <0.001 |
| | | | Proportions % (95%CI) | Proportions % (95%CI) | |
| Anovulatory cycles | | | 2.9% (2.7%-3.0%) | 2.5% (2.3%-2.6%) | <0.001 |
| Abnormal cycle lengths | | | 8.7% (8.4%-8.9%) | 8.0% (7.8%-8.2%) | <0.001 |
| Prolonged menses | | | 0.9% (0.8%-1.0%) | 1.0% (0.9%-1.1%) | 0.002 |
| | | | Users (%) | Users (%) | |
| Anovulatory cycles | No | | 16075 (88.9) | 16432 (90.9) | <0.001 |
| | Yes | | 2000 (11.1) | 1644 (9.1) | |
| Abnormal cycle lengths | No | | 12937 (71.6) | 13266 (75.13) | <0.001 |
| | Yes | | 5138 (28.4) | 4810 (26.6) | |
| Prolonged menses | No | | 17449 (96.5) | 17375 (96.1) | 0.04 |
| | Yes | | 626 (3.5) | 701 (3.9) | |

Abnormal cycle length: <24 days or >38 days; Prolonged menses: >8 days.

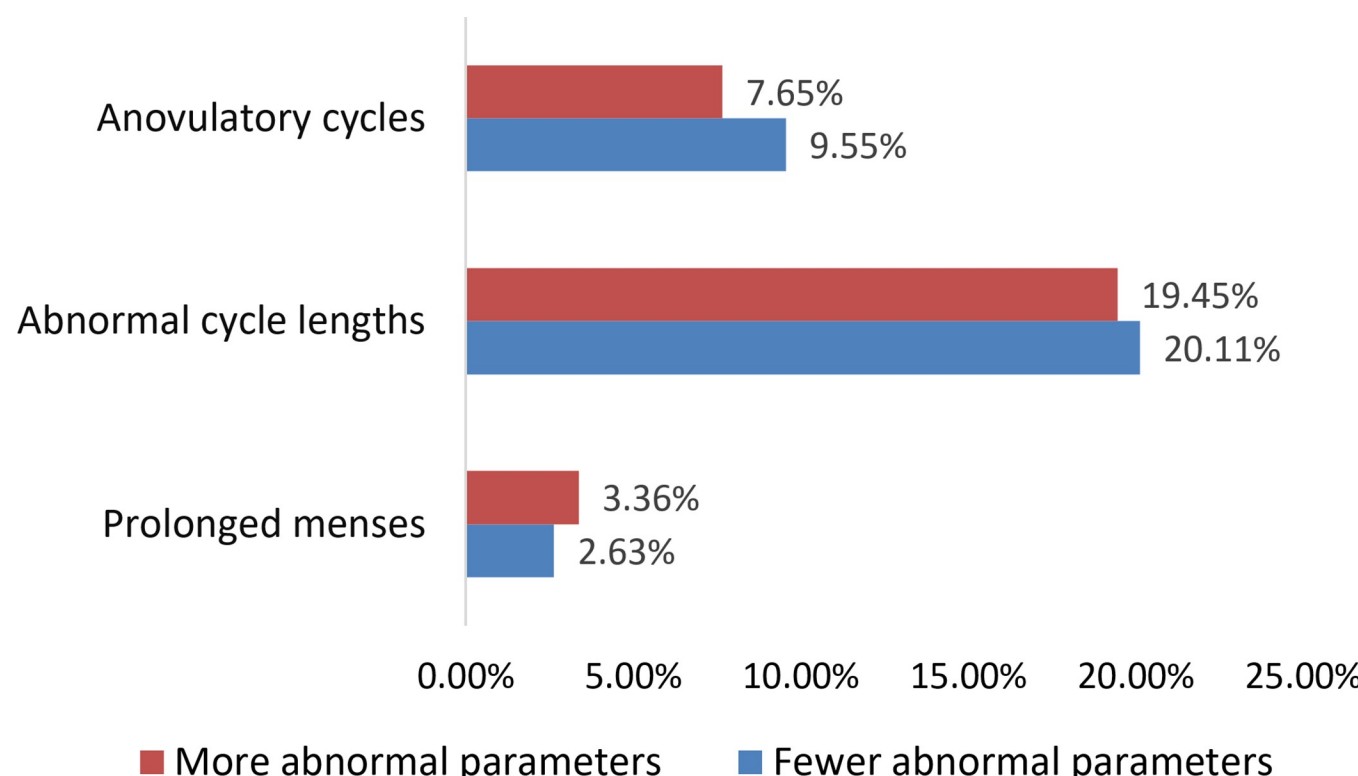

**Fig 1. Changes in the proportion of abnormal menstrual cycle parameters among users reporting more stress during than prior to the COVD-19 pandemic (n = 4,729).**

**Table 3. Users reporting more cycle abnormalities during the COVID-19 pandemic, by change in perceived stress (N = 10,293 users).**

|  | Less stress | No change | More stress | p-value |
|---|---|---|---|---|
|  | n = 2214 (%) | n = 3437 (%) | n = 4729 (%) |  |
| Anovulatory cycles (n = 776) | 175 (7.9) | 244 (7.1) | 357 (7.7) | 0.74 |
| Abnormal cycle lengths (n = 1942) | 413 (18.7) | 621 (18.2) | 908 (19.5) | 0.48 |
| Prolonged menses (n = 370) | 83 (3.8) | 130 (3.8) | 157 (3.4) | 0.55 |

**Table 4. Factors linked to more versus fewer cycle abnormalities during the COVID-19 pandemic (n = 18,076 users; N = 214,426 cycles).**

|  | Anovulatory cycles | | Abnormal cycle lengths | | Prolonged menses | |
|---|---|---|---|---|---|---|
|  | Fewer | More | Fewer | More | Fewer | More |
|  | aOR(95%CI) | aOR(95%CI) | aOR(95%CI) | aOR(95%CI) | aOR(95%CI) | aOR(95%CI) |
| **Age** |  |  |  |  |  |  |
| <25 | Ref | Ref | Ref | Ref | Ref | Ref |
| 25–34 | 0.49* | 0.46* | 0.71* | 0.79 | 0.98 | 0.80 |
|  | (0.36–0.66) | (0.32–0.64) | (0.55–0.92) | (0.59–1.06) | (0.50–1.93) | (0.46–1.40) |
| 35–44 | 0.38* | 0.44* | 0.65* | 1.17 | 1.24 | 0.97 |
|  | (0.27–0.53) | (0.31–0.65) | (0.49–0.87) | (0.86–1.60) | (0.60–2.54) | (0.53–1.76) |
| 45+ | 0.79 | 1.57* | 1.64* | 3.51* | 2.23 | 2.50* |
|  | (0.49–1.27) | (1.00–2.50) | (1.11–2.43) | (2.37–5.19) | (0.93–5.38) | (1.24–5.02) |
| **Education** |  |  |  |  |  |  |
| High school or less | Ref | Ref | Ref | Ref | Ref | Ref |
| Vocational training | 1.20 | 1.16 | 1.13 | 0.88 | 0.54 | 1.14 |
|  | (0.87–1.65) | (0.83–1.63) | (0.89–1.45) | (0.59–1.06) | (0.27–1.0) | (0.68–1.92) |
| University degree | 1.03 | 0.87 | 1.02 | 0.93 | 0.89 | 1.18 |
|  | (0.81–1.31) | (0.67–1.13) | (0.85–1.23) | (0.78–1.12) | (0.59–1.35) | (0.79–1.76) |
| PhD degree | 0.82 | 0.84 | 0.87 | 1.01 | 0.76 | 0.73 |
|  | (0.57–1.20) | (0.56–1.24) | (0.74–1.27) | (0.77–1.31) | (0.41–1.42) | (0.39–1.37) |
| **Relationship status** |  |  |  |  |  |  |
| In a relationship | 0.81 | 0.75* | 0.90 | 0.95 | 1.04 | 1.25 |
|  | (0.64–1.03) | (0.58–0.97) | (0.75–1.08) | (0.79–1.15) | (0.67–1.61) | (0.83–1.86) |
| Engaged or married | 0.74* | 0.72* | 0.92 | 0.88 | 0.94 | 0.99 |
|  | (0.57–0.96) | (0.54–0.95) | (0.75–1.12) | (0.72–1.08) | (0.59–1.52) | (0.64–1.53) |
| It's complicated | 1.09 | 0.94 | 1.00 | 0.93 | 0.51 | 1.18 |
|  | (0.74–1.62) | (0.60–1.45) | (0.73–1.38) | (0.67–1.30) | (0.19–1.35) | (0.62–2.27) |
| Single | Ref | Ref | Ref | Ref | Ref | Ref |
| **Children** |  |  |  |  |  |  |
| None | Ref | Ref | Ref | ref | Ref | Ref |
| One or more | 0.84 | 0.84 | 1.00 | 1.01 | 0.89 | 0.98 |
|  | (0.64–1.09) | (0.63–1.11) | (0.96–1.30) | (0.84–1.21) | (0.58–1.38) | (0.67–1.42) |
| **Subjective COVID-19 related stress** |  |  |  |  |  |  |
| Less | 0.91 | 1.02 | 1.11 | 1.10 | 1.10 | 0.92 |
|  | (0.74–1.12) | (0.81–1.29) | (0.91–1.25) | (0.93–1.28) | (0.77–1.56) | (0.67–1.27) |
| Unchanged | Ref | Ref | Ref | Ref | Ref | Ref |
| More | 0.98 | 1.07 | 1.08 | 1.12 | 0.82 | 0.88 |
|  | (0.83–1.16 | (0.88–1.29) | (0.95–1.23) | (0.98–1.28) | (0.60–1.12) | (0.68–1.14) |

* p<0.05.

stressors, we hypothesized that female menstrual cycle parameters would exhibit more abnormalities, inclusive of anovulation, during the COVID-19 pandemic. In this analysis of more than 200,000 cycles contributed by more than 18,000 app users with no recorded history of amenorrhea, we noted a statistically significant decrease in cycle length and increase in duration of menses at the population level, though these remained clinically unchanged at 29 and 4 days, respectively. At the individual user level, approximately 7.7% and 19.5% of users recorded more anovulatory cycles and abnormal cycle lengths following the pandemic, respectively; 3.4% recorded prolonged menses.

For some women, a new finding of amenorrhea, menorrhagia, or the inability to predict ovulation can be distressing and may lead to healthcare visits, laboratory workups, and imaging studies that may be unable to definitively determine an etiology. Contrary to expectation however, a larger proportion of mobile app users recorded fewer abnormalities during the pandemic than prior (9.6% and 20.1% recorded fewer anovulatory cycles and abnormal cycle lengths, respectively). Even when examining the cycle characteristics of women self-reporting more (32.0%) and markedly more (13.6%) stress following the pandemic, we did not observe any independent association of stress with cycle abnormalities. Rather than among women reporting high stress, we noted more abnormal cycles among women greater than the age of 45 years during the pandemic, which may suggest the stress-related sensitivity of the hypothalamic-pituitary axis during the perimenopausal state [14].

With respect to there being fewer abnormal parameters recorded during the pandemic, these findings may in part be explained by the sociodemographic characteristics of our sample population which may have provided some protection from both the physical and psychosocial effects of the pandemic. Our mobile app-using population was comprised primarily of healthy, college-educated women in their 30s, in relationships, and from majority Caucasian countries. Even during the first months of the pandemic, many individuals represented by the majority demographic in this sample were able to transition from commuting to working from home [15, 16], which based on analyses of global Twitter data, was perceived as a positive change among nearly 75% of individuals who additionally endorsed sentiments of trust, anticipation, and joy about the change during the pandemic [17]. Given the time and opportunity, some individuals may have engaged in exercise and better health habits, such as more regular sleep [18, 19], which taken together, might explain reductions in menstrual abnormalities during the pandemic.

In addition, we noted the protective effect of being in a relationship on the incidence of anovulation. While the COVID-19 pandemic may be exacerbating global gender inequality with more women transitioning to unemployment and reducing their work hours, women using the Natural Cycles app are primarily well-educated and from developed countries where one might expect more egalitarian gender role attitudes. In these households, gender roles may have been changing [20], with male partners spending more time at home and sharing both housework and childcare-related burdens [21]. These findings suggest that while the pandemic increased users' feelings of stress overall, that their health awareness, as well as their stable, protected working and living conditions, may have provided greater resilience and ability to control the situation [22].

In our multivariable logistic regression, we directly examined reports of marked increases in perceived stress during COVID-19, finding no association with abnormal menstrual parameters both before and after controlling for sociodemographic characteristics. Only 56.9% of users responding to the pandemic-related stress item, with the consequent possibility of selection bias; however, this sub-analysis still captured data from more than 10,000 users. For this population, general stress provoked by the COVID-19 pandemic and its preoccupations may not have been sufficient to elicit an adrenal and adaptive neuroendocrine response to acute

stress or chronic stress. We note as well that our chosen time points may not have been sufficiently long enough to observe stress-related physiologic effects which might not have been observed during the early months of the pandemic. Alternatively, our surface-level measurement of self-reported stress may have been insensitive to true distress or the behavioral determinants of stress-induced hypothalamic dysfunction [23]; our 2-item assessment of pandemic-related stress was likely biased towards inflated stress-reporting. While an ideal measure of change in stress would have repeated the question at two time points, we anticipated that user responses to the above two questions would still reflect perceived stress, as related to the pandemic. Yet even in studies where validated measures, such as the State-Trait Anxiety Inventory, are used, the relationship between the psychosocial status of women and aberrations in their menstrual cycles cannot always be demonstrated [24]. Future studies could collect serial stress assessments to evaluate the effects of chronic stress, as well as serum cortisol levels or diagnoses of COVID-19.

We also note that the proportion of cycles with prolonged menstruations increased during the pandemic, while the proportion of individuals with prolonged menses decreased, suggesting that those developing prolonged menses likely experienced recurrent prolonged menses. We are unable however, to make any inferences about the clinical significance of this finding as we did not quantify bleeding. Future versions of the app may expand its capabilities to help quantify bleeding and future analyses may examine the incidence of irregular spotting among users as a bothersome, early sign of menstrual irregularity.

The findings from this study are broadly limited by self-reported data. However, we believe that these data are collected from a group of individuals who are using the Natural Cycles app for its intended purpose of pregnancy prevention, rather than for the purposes of this research, such that these data are expected to be reliable. Mobile app-based reporting of menstrual data is preferred to conventional paper diary recording methods [25]. Further, previous publications on the menstrual cycle characteristics of more than 120,000 women using the Natural Cycles app to prevent or plan a pregnancy noted their individual contribution of approximately 9 cycles of data [26], suggesting user compliance with and acceptability of the method. Natural Cycles' recording of daily menstrual and basal body temperature diaries thus provided us with robust data for exploring female reproductive health and physiology in the setting of widespread socioenvironmental change—the COVID-19 pandemic. Future studies may be improved by the incorporation of real-time body temperature data collected by wearable devices, such as the Oura ring [27].

## Conclusion

The COVD-19 pandemic did not induce population-level changes to ovulation and menstruation among women using a mobile app to track menstrual cycles and predict ovulation. While some women experienced abnormalities during the pandemic, this proportion was smaller than that observed prior to the pandemic. As most app users in this study were well-educated women over the age of 30 years, and from high-income countries, their experience of the COVID-19 pandemic may differ in ways that limit the generalizability of these findings.

## Acknowledgments

Dr. Brian T. Nguyen would like to acknowledge the support of his wife, Amy Li, whose predictable menstrual cycles led to the arrival of their newborn daughter, Charlotte Li Nguyen, during the COVID-19 pandemic.

## Author Contributions

**Conceptualization:** Brian T. Nguyen, Anita L. Nelson, Jack T. Pearson, Hana R. Reissner, Anita Kraker von Schwarzenfeld, Juan Acuna.

**Data curation:** Brian T. Nguyen, Anita L. Nelson, Jack T. Pearson, Eleonora Benhar Noccioli, Anita Kraker von Schwarzenfeld, Juan Acuna.

**Formal analysis:** Brian T. Nguyen, Raina D. Pang, Juan Acuna.

**Investigation:** Brian T. Nguyen, Anita L. Nelson, Jack T. Pearson, Eleonora Benhar Noccioli, Juan Acuna.

**Methodology:** Brian T. Nguyen, Raina D. Pang, Anita L. Nelson, Jack T. Pearson, Eleonora Benhar Noccioli, Hana R. Reissner, Anita Kraker von Schwarzenfeld, Juan Acuna.

**Project administration:** Brian T. Nguyen, Eleonora Benhar Noccioli.

**Software:** Eleonora Benhar Noccioli.

**Supervision:** Raina D. Pang, Anita L. Nelson, Juan Acuna.

**Visualization:** Brian T. Nguyen.

**Writing – original draft:** Brian T. Nguyen, Hana R. Reissner.

**Writing – review & editing:** Brian T. Nguyen, Raina D. Pang, Anita L. Nelson, Jack T. Pearson, Eleonora Benhar Noccioli, Hana R. Reissner, Anita Kraker von Schwarzenfeld, Juan Acuna.

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
