## [Decision Letter · Decision Letter 0]

3 Jun 2021

PONE-D-21-15151

Detecting variations in ovulation and menstruation during the COVID‐19 pandemic, using real‐world mobile app data

PLOS ONE

Dear Dr. Nguyen,

Thank you for submitting your manuscript to PLOS ONE. After careful consideration, we feel that it has merit but does not fully meet PLOS ONE’s publication criteria as it currently stands. Therefore, we invite you to submit a revised version of the manuscript that addresses the points raised during the review process.

The manuscript presents some interesting data.

Before re-evaluating the possibility of publication, it is necessary that the authors provide feedback to the comments of the reviewers.

I also ask the authors to provide more details on the method the app uses to determine ovulation.

There is a comment the authors made claiming that their method has a Pearl index of 6.9, which they claim is comparable to hormonal contraceptives. This statement seems to be an advertisement for their app, and, as it is not strictly relevant to the content of the article, I think that has to be removed from the paper.

We look forward to receiving your revised manuscript.

Kind regards,

Alessio Paffoni, PhD

Academic Editor

PLOS ONE

Journal Requirements:

2. Please include your tables as part of your main manuscript and remove the individual files. Please note that supplementary tables (should remain/ be uploaded) as separate "supporting information" files

3.We note that you have indicated that data from this study are available upon request. PLOS only allows data to be available upon request if there are legal or ethical restrictions on sharing data publicly. For information on unacceptable data access restrictions, please see http://journals.plos.org/plosone/s/data-availability#loc-unacceptable-data-access-restrictions.

4.Thank you for stating the following in the Financial Disclosure section:

"The authors received no specific funding for this work."

We note that one or more of the authors are employed by a commercial company: Natural Cycles

Reviewers' comments:

Reviewer's Responses to Questions

**Comments to the Author**

1. Is the manuscript technically sound, and do the data support the conclusions?

Reviewer #1: No

Reviewer #2: Yes

Reviewer #3: Yes

2. Has the statistical analysis been performed appropriately and rigorously? 

Reviewer #1: Yes

Reviewer #2: Yes

Reviewer #3: Yes

3. Have the authors made all data underlying the findings in their manuscript fully available?

Reviewer #1: Yes

Reviewer #2: Yes

Reviewer #3: Yes

4. Is the manuscript presented in an intelligible fashion and written in standard English?

Reviewer #1: Yes

Reviewer #2: Yes

Reviewer #3: Yes

5. Review Comments to the Author

Reviewer #1: Detecting variations in ovulation and menstruation during the COVID-19 pandemic, using real-world mobile app data.

The authors present a very large data set, collected through the app Natural Cycles, which provides a firm basis for real world conclusions. The data herein details changes in menstrual functionality amongst women during the COVID pandemic, comparing to data pre pandemic. The aim and relevance of the data collected is clear and necessary, however there are queries amongst how the data is collected through the app to determine ovulation. This needs to be clarified in more detail within the manuscript. Please see specific comments below.

Abstract: The first sentence of the abstract needs to be reworded or include more commas, as it stands this is not a very clear start to the manuscript.

The authors state they restricted the data set to regular cycling women yet do not include the clear, readily used, definition of eumenorrheic. The authors should detail which definition they have used in the abstract. Further comments on this below.

Ethics statement: The authors have stated that it was clarified that this was a human exempt study, could the authors make it explicitly clear that whilst this study was exempt that ethical approval was still sought?

Introduction

Line 123: ‘The events set…’ this depicts that it was the lockdowns that caused the concerns about the pandemic alone, in fact there were many different situations and events on going across the globe and collectively that raised the levels of concern. I would suggest re structure to ensure this comes through in the text.

Line 128: …’ability to reproduce’ I can see the authors need to introduce this concept to the text at this point as to the purpose of their manuscript however the placement of this here does not fit. The ability to reproduce was certainly not an immediate outcome of the pandemic nor do we yet have the scientific data to suggest this is a consequence of the long COVID for example. I would suggest either removing or expanding here on how the pandemic directly affects the ability of people to reproduce.

Line 139: …’transmitted generationally’ please provide a reference here that demonstrates the affects of stress on the menstrual cycle can be transmitted between generations.

Line 147: typo.

Line 157-161. The last sentence of this section needs to be re worded, it is not coherent in its current format. This sentence is making quite assumptive statements too I would suggest being more direct in making your point and include references.

Line 177: Reasonably certain? Can the authors provide clarification or quantification to this?

Methods:

The main issue here is the lack of clarity on determining ovulation. This is one of the main aims/outcomes of the study yet there are no specific details on how ovulation was detected through the app. There are some suggestions of basal body temperature, but no detail. This is imperative to be provided. Were all users testing temperature at same time of day? Were they using the same device? Did they all test the temperature regularly? If not how often? Without such information it makes the data seem quite questionable as to how the results can be deemed valid and/or appropriate.

Line 239. The definition of normal cycles often used in research is 21-35days long, please could the authors explain why they chose 24 days as the shortest cycle to include?

Line 254-255: The covariates stated are all reasonable but again no information of the questions as to how this was collected or categorised. Please provide such detail.

Results

The display of data is very poor. Figure 1 in particular has different coloured text/font/size. The key does also not correspond with the data being shown. Please revise thoroughly.

The symbols used in the tables to display significance, it is not clear which category this is referring to compared to which other category. Please revise.

Table IV is also quite confusing as to whether this is referring to categories of stress or not. I would suggest changing to a clearer display method or altering the table/better legends.

Line 293: include the significance value.

Lines 299-301 The authors state a significant change in cycle length yet looking at the numbers this changes by 0.5 day, the authors need to clarify the length of cycles users inputted as it seems somewhat strange that a user would have a cycle length of 29.5 days.

Line 301: Please include here the number of ‘average incidence’ of ovulation.

Line 319: Indicates that more variable cycles came from older participants, yet the authors have not discussed later that this could be linked to peri menopause as well. I would suggest mentioning this as a possibility within the discussion.

Discussion:

As per above there needs to be more a more detailed discussion of the possible limitations which should include the ovulation reliability, possibility of peri menopause, as well as the possible bias to the data set as already included. I would suggest a specific limitation section should be included as the whole basis of the data collection is reliant upon people taking their own ovulation testing and truthfully inputting the data. There are many studies that have discussed the inaccuracies of self-report data but none have been acknowledged or references. Please include such.

Line 368: In a May 2020…Please re word.

A clearer conclusion/future work section needs to be included, the manuscript ends somewhat abruptly.

Reviewer #2: This paper is quite interesting, and addresses a very commonly asked question by women during the pandemic. The authors do discuss the limitations of their paper, as far as the study group (Highly educated, presumably well off, young women, in general) but they have a considerable data base. And their negative findings are very interesting. As an ancillary question, which has nothing to do with the current study: I would ask, perhaps from a sociologist or psychologist, why this group of women in 2019 (going back to their control group) were experiencing so much stress? If I read the numbers correctly, almost half (46.2%) of the women studied stated that they were either very or extremely stressed in 2019. Why? But I think the study as is is quite succinct and helpful.

Reviewer #3: This is a very important and time-sensitive topic that has received a lot of media attention for anecdotal report of menstrual changes during the COVID-19 pandemic. It is great to have data to start to answer some of these questions. This manuscript contains valuable information but needs editing and some clarifications to make it suitable for publication.

Overall comments:

1. The entire paper is too long and wordy. It would benefit from editing, especially in the Abstract, Introduction, and Discussion. More details below.

2. Better clarification is needed about the definition of anovulation. While it appears that you are using definitions of ovulation/anovulation as defined by a confidential algorithm, it is confusing because regular menses and cycle length are typically used as proxies for ovulatory cycles, and yet anovulation and abnormal cycle length and prolonged menses are presented as completely different concepts here.

3. Similarly, you seem to imply that regular menstrual cycles was an inclusion criterion but then report anovulatory cycles in this cohort. I believe that you are excluding amenorrhea or “very irregular” cycles but need to clarify these concepts and definitions better.

Abstract:

-Much too long. The abstract should not be longer than 300 words and this seems much longer.

-Subheadings are typically Background, Methods, Results, Conclusions. There is no summary/conclusions in your abstract.

-Limitations are not included in the abstract.

-Have never seen an Extended Abstract before. Was this specifically requested? If not, please remove. Your abstract should speak for itself and lead readers to read the entire paper.

Introduction:

-Too long, should not be more than 2-3 paragraphs. Needs editing.

-Last paragraph should be moved to Methods.

-Section should end with a clear statement of the objective of this study.

Methods

-The Inclusion/Exclusion criteria section is long-winded and confusing. Needs to be tightened up and clearer. The most confusing part is the exclusion of those with any cycle >90 days “to account for any undiagnosed amenorrhea or ovulatory dysfunction.” This needs to be clarified. I believe what you mean is that you excluded anyone with secondary amenorrhea, as defined by any cycle >90 days. You did not exclude anyone with irregular cycles, as those would have been anyone with cycles <24 or >38 days, per the FIGO definitions you cite on the next page. As such, the line in the Outcomes section about “… the proportion of users experiencing anovulation, abnormal cycle length, and prolonged menstruation among regularly cycling app users …” seems contradictory. The phrase “regularly cycling app users” should be eliminated throughout the manuscript, as you are not necessarily including those with regular cycles, just those who are not amenorrheic.

-Why were women with endometriosis excluded? What is the evidence for endometriosis affecting menstrual regularity?

-I assume that women using this app are not on any hormonal birth control method, but that should be explicitly stated.

-The last 3 sentences of the Inclusion/Exclusion section are redundant and should be removed.

-Outcome section: Please clarify the line “which may be supplemented by positive urinary LH tests.” Was this done for all women?

Results

-Be consistent in the text and tables about the numbering system for tables. You use Arabic numbers (e.g. 1, 2) in the text but Roman numerals (e.g. I, II) for the tables themselves.

-If you state that there is a “significant” difference in the text and include numbers/percentages for the groups, then you should also include p-values in the text. This is very inconsistent.

-There is no reason to include both Figure 1 and Table 3, as both give the same information. I think it is presented better in the figure.

Discussion

-This is too long. The purpose of the Discussion is to summarize and interpret your most significant findings and compare them to prior data. You should not rehash all the Results.

-You take almost 2 full pages at the end of the Discussion to make the same point. 1) This should be significantly shortened. 2) You must also consider the possibility that routine stress, as experienced by many during COVID-19, is different than the stress experienced during, for example, war and that COVID-19 might not have caused significant physiologic stress and might not have had any affect on menses and ovulation.

-The final paragraph is out of place as the concluding paragraph. Shorten to 1-2 sentences as additional limitations and add a true Conclusions final paragraph to the paper that summarizes the importance of your findings and next steps.

Tables

-Consistent numbering, per comment above

-Why does Table 2 have an * in 1 place? What does this mean? Needs to be defined under table.

6. PLOS authors have the option to publish the peer review history of their article (what does this mean?). If published, this will include your full peer review and any attached files.

Reviewer #1: No

Reviewer #2: **Yes: **Mary Jane Minkin, MD

Reviewer #3: **Yes: **Beth I. Schwartz, MD

---

## [Author Response · Author response to Decision Letter 0]

25 Jun 2021

Please see the separately attached reviewer response for a point-by-point address of each correction.

---

## [Decision Letter · Decision Letter 1]

13 Jul 2021

PONE-D-21-15151R1

Detecting variations in ovulation and menstruation during the COVID‐19 pandemic, using real‐world mobile app data

PLOS ONE

Dear Dr. Nguyen,

Thank you for submitting your manuscript to PLOS ONE. After careful consideration, we feel that it has merit but does not fully meet PLOS ONE’s publication criteria as it currently stands. Therefore, we invite you to submit a revised version of the manuscript that addresses the points raised during the review process.

Both reviewers agree that the text has improved. However, some aspects remain that require attention by the authors as specified below.

We look forward to receiving your revised manuscript.

Kind regards,

Alessio Paffoni, PhD

Academic Editor

PLOS ONE

Reviewers' comments:

Reviewer's Responses to Questions

**Comments to the Author**

1. If the authors have adequately addressed your comments raised in a previous round of review and you feel that this manuscript is now acceptable for publication, you may indicate that here to bypass the “Comments to the Author” section, enter your conflict of interest statement in the “Confidential to Editor” section, and submit your "Accept" recommendation.

Reviewer #1: (No Response)

Reviewer #3: (No Response)

2. Is the manuscript technically sound, and do the data support the conclusions?

Reviewer #1: Yes

Reviewer #3: Yes

3. Has the statistical analysis been performed appropriately and rigorously? 

Reviewer #1: Yes

Reviewer #3: No

4. Have the authors made all data underlying the findings in their manuscript fully available?

Reviewer #1: Yes

Reviewer #3: Yes

5. Is the manuscript presented in an intelligible fashion and written in standard English?

Reviewer #1: Yes

Reviewer #3: Yes

6. Review Comments to the Author

Reviewer #1: Thank you to the authors for address the majority of the queries put forward by both reviewers. The manuscript provides a lot more detail in its current format. I do still have some concerns/queries that need to be addressed which I have provided in the comments below (see marked comment 2).

Original comments: Methods: The main issue here is the lack of clarity on determining ovulation. This is one of the main aims/outcomes of the study yet there are no specific details on how ovulation was detected through the app. There are some suggestions of basal body temperature, but no detail. This is imperative to be provided. Were all users testing temperature at same time of day? Were they using the same device? Did they all test the temperature regularly? If not how often? Without such information it makes the data seem quite questionable as to how the results can be deemed valid and/or appropriate.

Original response: We have provided more detail about how basal body temperatures are recorded as follows: “Anovulatory cycles were defined according to the Natural Cycles app’s proprietary algorithm. The algorithm identifies ovulation retrospectively based on the first day of menstruation and basal body temperatures, which may be supplemented by positive urinary LH tests. Basal body temperatures are recorded each morning using a thermometer sensitive to the hundredth place, and with measures excluded if the user reports any illness or changes in sleep that might influence basal temperatures. Users of Natural Cycles record basal temperatures for approximately 70% of the days. To reduce the risk of misidentifying ovulations, the algorithm reports ovulation by rising basal body temperature only if the average temperature from three consecutive calendar days is greater than the woman’s follicular phase average and her baseline average across all data entries, as well as consistent with her luteal phase average [15]. If no temperature rise is observed and the data quality and quantity is deemed sufficient, the cycle is flagged as anovulatory. Cycles with low data quality or many missing data points are excluded by the analysis.”

Comment 2: Thank you for providing this important detail. Could the authors please clarify the distribution of reporting further. The authors state that basal temperatures are recorded approximately 70% of the days, it is imperative to provide detail on this, what days were reported more than others? Did it vary between participants using the app? If certain days were continually omitted in some cases then this would contribute somewhat to the outcomes of the analysis.

Comment 2: The authors also state that cycles will low data quality or any missing data points are excluded by the analysis. Again the detail is required here. What was the authors definition of low quality data (is this linked to the statement prior to this or something else?) and provide exactly how many missing data points that then meant the cycle was excluded.

Original comments: Ref (line 239) is from 2007 there should be more recent ref used to support the use of 24 days.

Original response: Line 239. The definition of normal cycles often used in research is 21-35 days long, please could the authors explain why they chose 24 days as the shortest cycle to include? • We used criteria from FIGO (https://pubmed.ncbi.nlm.nih.gov/17362717/), which are based on more recent population estimates at the 5th and 95th percentile. Had we used 21 days, we would have significantly underestimated frequent menstruations and potentially been less likely to find a difference.

Comment 2: The reference the authors have provided is dated 2007, almost 14 years ago. There are many updated references that should be provided for clarification of their use of 24 days as the shortest cycle length to include. Please update.

Comment 2: Figure 1 is still very poor. I would suggest removing the background lines. Ensure all text is black font to align with the manuscript.

Original comment: Lines 299-301 The authors state a significant change in cycle length yet looking at the numbers this changes by 0.5 day, the authors need to clarify the length of cycles users inputted as it seems somewhat strange that a user would have a cycle length of 29.5 days.

Original Response: The cycle lengths are averaged from the number of cycles that are contributed over the study periods, which is why the averages are fractions of whole numbers.

Comment 2: My comment has still not been fully addressed here I apologise if I was not clear originally. My query is regarding the reporting of the cycle through the app. Generally, participants would report their cycle lengths in whole days. Is it feasible for a participants/users cycle to be reported for example as 29.5 days if they reported precisely in the day when bleeding started/ stopped and began again for the next cycle? Is this something that is feasible through the app? I fully appreciate and understand that the 0.5 increase would be an average.

Original comment: Discussion: As per above there needs to be more a more detailed discussion of the possible limitations which should include the ovulation reliability, possibility of peri menopause, as well as the possible bias to the data set as already included. I would suggest a specific limitation section should be included as the whole basis of the data collection is reliant upon people taking their own ovulation testing and truthfully inputting the data. There are many studies that have discussed the inaccuracies of self-report data but none have been acknowledged or references. Please include such.

Original response: We note that the reviewer is very focused on a biased critique of the app being unable to accurately predict ovulation. We note that ovulation in all the possible ways in which it can be measured non-invasively are soft markers. Even the gold standard of using ultrasound cannot determine if a ruptured follicle has released an egg. To suggest that our research would be improved via a study that demanded transvaginal ultrasound and routine LH kit testing would be infeasible at our sample size. Ovulation prediction is limited, which is why our investigation additionally includes other menstrual parameters aimed at increasing our ability to detect changes related to the pandemic. The reviewer’s comments do not acknowledge that this data set is one of largest to collect these data and that any data collection performed in a clinical setting would be subject to even greater limitations. Any paper-collected self-report would be even less reliable and more likely to reflect recall bias, which our methods avoid by requiring daily data entry. Abiding by the reviewer’s request would turn a strength of our methodology into a weakness.

Original response: That being said, we have added the following broad limitation acknowledgement in a separate limitations section as follows: “The findings from this study are broadly limited by self-reported data. However, we believe that these data are collected from a group of individuals who are using the Natural Cycles app for its intended purpose of pregnancy prevention, rather than for the purposes of this research, such that these data are expected to be reliable.”

Comment 2: I have already acknowledged the large data set in the very first comment to the authors. Nor have I ever suggested an alternative more invasive method be used such as transvaginal US. I have focused on the ovulation due to the previous lack of detail provided, therefore making it difficult to understand as a reader how an app would predict ovulation. The information that has been added has certainly now helped the manuscript. However, it is still important to acknowledge the limitations, as with all original research. The reporting of basal body temperature, despite being daily, is still reliant on self-report as it is trusting participants will not falsify any data, as is the same with all self report data. It does not mean to say your data is not reliable it is just something that cannot be guaranteed. Therefore, I would suggest adding to your acknowledgment of this with a reference regarding the reliability of self-report data.

Reviewer #3: This paper is significantly improved but still needs some edits to make it suitable for publication. Specific comments below.

Abstract: Much better! Only comment is that I would reword the last sentence both here and in the Discussion to read "... their experiences of the COVID-19 pandemic might differ ..."

Introduction: Significantly improved. You still need to end this section with a clearer sentence on the objective of the study. Most papers end with a line that literally states "The objective of this study is to ..."

Methods:

-Page 6, Lines 123-125: This seems unnecessary and like an ad for the app.

-Page 6, Inclusion/Exclusion Criteria: Were cycles of users who became pregnant during use included?

-Page 8, Lines 171-172: This was stated earlier and is redundant.

Results:

-Relationship status in Table 1 seems to need a +.

-It is unusual to present 95% CI for means, instead of SD. Please present that in the Methods and address why.

-I continue not to think that both Figure 1 and Table 3 are both necessary, especially as there are no differences and this can be just addressed in the text.

-Page 13, Lines 243-247: Creation of the model should be in the Methods section. However, I do not see that you actually created a model. Table 4 is just a list of significant associated risk factors and is not a model.

Discussion:

-My biggest concern is that you take 1.5 pages of a 2.5 page discussion on a theory that defends your lack of expected results. It is ok to mention this but not to take up this much space on it. Instead, spend some time discussing the implications of your actual results.

-Page 16, Lines 284-289: I don't understand how this is relevant.

-Page 17, Lines 312-315: Too much rehashing. Would rewrite as "Only 56.9% of users .... bias; however, this sub-analysis ..."

-Page 18, Limitations: You could consider the possibility that the chosen time points were not enough to capture changes in menses and ovulation, as it might take more time to see the physiologic effects on stress, as COVID-19 did not start until mid-March 2020 and most thought it would be very temporary at that time.

Conclusion:

-Page 19, Line 350: Per my comment in the Abstract, would change this to "may"

-Suggest ending with a line "Future research is needed to ..."

7. PLOS authors have the option to publish the peer review history of their article (what does this mean?). If published, this will include your full peer review and any attached files.

Reviewer #1: No

Reviewer #3: **Yes: **Beth I. Schwartz, MD

---

## [Author Response · Author response to Decision Letter 1]

2 Sep 2021

Please see the separately attached reviewer response for a point-by-point address of each correction.

---

## [Decision Letter · Decision Letter 2]

20 Sep 2021

PONE-D-21-15151R2Detecting variations in ovulation and menstruation during the COVID‐19 pandemic, using real‐world mobile app dataPLOS ONE

Dear Dr. Nguyen,

Thank you for submitting your manuscript to PLOS ONE. After careful consideration, we feel that it has merit but does not fully meet PLOS ONE’s publication criteria as it currently stands. Therefore, we invite you to submit a revised version of the manuscript that addresses the points raised during the review process.

The revised manuscript has been appreciated. Minor revisions are needed at this point. I believe that table 3 and Figure 1 can remain in the final text.

We look forward to receiving your revised manuscript.

Kind regards,

Alessio Paffoni, PhD

Academic Editor

PLOS ONE

Journal Requirements:

Additional Editor Comments (if provided):

Reviewers' comments:

Reviewer's Responses to Questions

**Comments to the Author**

1. If the authors have adequately addressed your comments raised in a previous round of review and you feel that this manuscript is now acceptable for publication, you may indicate that here to bypass the “Comments to the Author” section, enter your conflict of interest statement in the “Confidential to Editor” section, and submit your "Accept" recommendation.

Reviewer #1: (No Response)

Reviewer #3: (No Response)

2. Is the manuscript technically sound, and do the data support the conclusions?

Reviewer #1: Yes

Reviewer #3: Yes

3. Has the statistical analysis been performed appropriately and rigorously? 

Reviewer #1: Yes

Reviewer #3: Yes

4. Have the authors made all data underlying the findings in their manuscript fully available?

Reviewer #1: Yes

Reviewer #3: Yes

5. Is the manuscript presented in an intelligible fashion and written in standard English?

Reviewer #1: Yes

Reviewer #3: Yes

6. Review Comments to the Author

Reviewer #1: Thank you for addressing my points.

The only remaining point is regarding the statement 'many missing data points' when discussing the data that is included for final analysis. Please could the authors change this precise statement to make more quantifiable and perhaps in relation to the '10 temperature readings' that are stated to have been required earlier. This is my only remaining query.

Reviewer #3: Thank you for your thoughtful and substantial edits. The paper is now in very good shape! Only 2 comments:

1) Would recommend slight edits to the Limitations section. This seems to be a continuation of the paragraph above, rather than a clear statement of the limitations of the study, which does not clearly begin until >1 paragraph later. You could even just remove the Limitations subheading, and it would read better.

2) I continue to note that I do not think that both Figure 1 and Table 3 are both necessary, especially as there are no differences and this can be just addressed in the text. However, this should not stop the paper from being published in its current form.

7. PLOS authors have the option to publish the peer review history of their article (what does this mean?). If published, this will include your full peer review and any attached files.

Reviewer #1: No

Reviewer #3: **Yes: **Beth I. Schwartz, MD

---

## [Author Response · Author response to Decision Letter 2]

21 Sep 2021

Dear editors and reviewers, 

Thank you for your favorable review of our manuscript, “Detecting variations in ovulation and menstruation during the COVID-19 pandemic, using real-world mobile app data.” We acknowledge the reviewers’ comments and have made revisions as appropriate with responses below. Please find our review responses highlighted in yellow.

Reviewer #1: Thank you for addressing my points. The only remaining point is regarding the statement 'many missing data points' when discussing the data that is included for final analysis. Please could the authors change this precise statement to make more quantifiable and perhaps in relation to the '10 temperature readings' that are stated to have been required earlier. This is my only remaining query. 

Thank you for your query about the ambiguity in our manuscript. However, the explanation of how cycles are excluded is not as simple to explain as to provide a cutoff or threshold value. The quantity of missing data that is acceptable is a function as well of the stability of the user’s data as well. To help the reader understand this and reduce the ambiguity in the manuscript we have revised as follows (we have removed mention of “many missing data points”:

“If no temperature rise is observed and the data quality and quantity is deemed sufficient, the cycle is flagged as anovulatory. Of note, users with stable measurements (e.g., small day-to-day variations in the same cycle phase), require fewer data points for the Natural Cycles algorithm to draw conclusions about changes in the basal body temperature. Cycles with low data quality (e.g., high day-to-day temperature variability) or data that is insufficient to detect or exclude an increase in the basal body temperature are excluded by the analysis.”

Reviewer #3: Thank you for your thoughtful and substantial edits. The paper is now in very good shape! Only 2 comments: 

1) Would recommend slight edits to the Limitations section. This seems to be a continuation of the paragraph above, rather than a clear statement of the limitations of the study, which does not clearly begin until >1 paragraph later. You could even just remove the Limitations subheading, and it would read better. 

- Thank you for your comment, for which we’ve removed the “Limitations” subheading to help with the flow of the discussion section. 

2) I continue to note that I do not think that both Figure 1 and Table 3 are both necessary, especially as there are no differences, and this can be just addressed in the text. However, this should not stop the paper from being published in its current form. 

- Per editor, Fig 1 and Table 3 may remain in final text.

---

## [Editor Report · Decision Letter 3]

24 Sep 2021

Detecting variations in ovulation and menstruation during the COVID‐19 pandemic, using real‐world mobile app data

PONE-D-21-15151R3

Dear Dr. Nguyen,

We’re pleased to inform you that your manuscript has been judged scientifically suitable for publication and will be formally accepted for publication once it meets all outstanding technical requirements.

Kind regards,

Alessio Paffoni, PhD

Academic Editor

PLOS ONE

---

## [Editor Report · Acceptance letter]

12 Oct 2021

PONE-D-21-15151R3 

Detecting variations in ovulation and menstruation during the COVID-19 pandemic, using real-world mobile app data 

Dear Dr. Nguyen:

I'm pleased to inform you that your manuscript has been deemed suitable for publication in PLOS ONE. Congratulations! Your manuscript is now with our production department. 

Kind regards, 

on behalf of

Dr. Alessio Paffoni 

Academic Editor

PLOS ONE